# Correlation between Structural and Functional Changes in Patients with Raised Intraocular Pressure Due to Graves’ Orbitopathy

**DOI:** 10.3390/diagnostics14060649

**Published:** 2024-03-20

**Authors:** Freja Bagatin, Ante Prpić, Jelena Škunca Herman, Ognjen Zrinšćak, Renata Iveković, Zoran Vatavuk

**Affiliations:** Department of Ophthalmology, Sestre Milosrdnice University Hospital Center, 10000 Zagreb, Croatia; ap.prpic@gmail.com (A.P.); jskuncaherman@gmail.com (J.Š.H.); ozrinscak@yahoo.co.uk (O.Z.); renata.ivekovic@kbcsm.hr (R.I.); zo.vatavuk@gmail.com (Z.V.)

**Keywords:** Graves’ orbitopathy, ocular hypertension, risk factors

## Abstract

This study explores the complication of secondary intraocular pressure (IOP) elevation and consequent glaucoma development in Graves’ orbitopathy (GO), an autoimmune disorder associated with hyperthyroidism. Utilizing Octopus 900 visual field testing and optical coherence tomography (OCT), the research established correlations between functional and structural changes in optic nerve regions in patients with GO and patients with GO with elevated IOP (GO IOP) groups. A comparison with primary open-angle glaucoma (POAG) was conducted in a cohort of 182 subjects. The study identifies optic nerve head parameters that effectively differentiate changes in GO and GO IOP groups. In the GO group, the strongest correlation between structural and functional changes was observed in sector 7, while in the GO IOP group, it was in sectors 1 and 7. For POAG, correlation was found in six sectors. Elevated IOP in GO correlates with structural and functional impairments similarly to early glaucoma. Risk factors for GO-related elevated IOP included older age, longer duration of thyroid disease, and higher anti-thyroglobulin values. The study highlights the significance of regular IOP measurements, visual field assessments, and OCT examinations in GO patients. Early antiglaucoma intervention is warranted when characteristic structural and functional changes and/or risk factors are identified.

## 1. Introduction

Graves’ ophthalmopathy or orbitopathy (GO) is an autoimmune condition caused by Graves’ disease. The autoimmune aspect of GO involves the immune system mistakenly attacking the tissues around the eyes. The underlying cause of GO is thought to be the production of antibodies (such as thyroid-stimulating immunoglobulins) that target both the thyroid gland and the tissues around the eyes. These antibodies play a role in the development and progression of the eye-related symptoms affecting eye muscles and tissues seen in GO [1]. A secondary rise in intraocular pressure (IOP) in GO and a consequent loss of retinal ganglion cells and axons of the retina as a complication arise from a different mechanism than in primary open-angle glaucoma (POAG) [2]. The estimated prevalence of ocular hypertension (OHT) in subjects with GO varies considerably from 3.1 to 24%, and that of open angle glaucoma from 0.8 to 13% [3]. There are a number of mechanisms by which thyroid disorders can cause the development of glaucomatous damage. Possible causes of elevated IOP include elevation of episcleral venous pressure as a consequence of orbital congestion, increase in retrobulbar pressure, restriction and compression of the eyeball by the contraction of enlarged and fibrotic extraocular muscles, glycosaminoglycan deposition in the trabecular meshwork, and genetic predisposition to thyroid disease and glaucoma [4,5,6].

The study aims to correlate functional visual field changes with structural changes in the optic nerve using Polar Analysis, a tool of the Octopus 900 visual field, and optical coherence tomography (OCT). Although the structural–functional (s-f) correlation has been investigated in POAG [7,8], it has rarely been examined in GO.

The localization and the extent of s-f changes in patients with GO, GO with raised IOP, and POAG will be compared to determine if the elevation of IOP in GO causes significant changes and if hypotensive therapy is always necessary. The formation of an objective s-f map that could be used to improve the ability of clinicians to diagnose, monitor, and treat glaucoma in GO patients will be explored. Furthermore, the study will analyse risk factors for GO and their potential role in elevation of IOP.

## 2. Materials and Methods

The study was conducted following the Declaration of Helsinki and approved by the Ethics Committee of Clinical Hospital Centre “Sestre Milosrdnice” (protocol code EP-5992/17-1, 6 April 2017). GO diagnosis was based on clinical, computed tomography (CT), and endocrinologic findings, classified using the index of disease activity based on clinical signs (CAS), the severity of the disease by NOSPECS and EUGOGO classification [9]. Severe GO cases impeding visual field recording were excluded. The elevated IOP group included subjects with IOP > 21 mmHg in three subsequent measurements. POAG was defined by optic nerve changes, glaucomatous visual field defects, and IOP > 21 mmHg in the presence of an open angle. Only mild to moderate POAG patients were included [10]. Exclusions comprised age < 18, best corrected visual acuity (BCVA) < 0.8, refractive error > ±5 diopters, astigmatism > ±2 diopters, previous surgeries and procedures on the eye, cloudiness in the optical media of the anterior eye segment (cornea, anterior chamber, lens), closed angle on gonioscopy, use of steroids in the last 6 months and dysthyroid opticoneuropathy (DON). Ophthalmological tests-best corrected visual acuity (BCVA) with Snellen charts, IOP (Goldmann applanation tonometry), central corneal thickness (CCT) on Tomey em-4000 (Tomey corporation, Nagoya, Japan), Mourits exophthalmometry, gonioscopy, Octopus 900 visual field dynamic strategy/G2 pattern (Haag-Streit, Köniz, Switzerland) recordings, and Cirrus HD-OCT (Carl Zeiss Meditec, Inc., Dublin, CA, USA) measurements were conducted. OCT values were based on the referral values in OCT interpretation guide. Each subject’s visual field was recorded consecutively three times at a one-month interval to reduce the number of false-positive and false-negative results. The third finding of the visual field was analysed. The Octopus Polar Analysis was divided into 12 sectors corresponding to the clock-hour retinal nerve fibre layer (RNFL) thickness sectors obtained by OCT. The calculated damage in decibels (dB) was correlated with the structural damage on OCT in the same region in microns (μm). In Polar analysis, local defects are displayed as red lines and scaled with rings for 10, 20, and 30 dB deviation in 12 equal sectors. Each line represents the defect of each test location at the nerve fibre angle at the optic disc. The orientation is vertically mirrored compared to the representation in the visual field (grey scale) to correspond to structural changes in the head of the optic nerve (Figure 1A). The severity of damage within the sector is not expressed numerically, and therefore, the total damage in dB in each of the 12 sectors for each patient was calculated manually. Damage in each sector in dB was compared to structural damage on OCT in the same region in μm and the possible correlation of these two procedures for each eye was calculated (Figure 1). For each subject with GO, CAS, NOSPECS, and EUGOGO classification [11], thyroid hormones (fT3 and fT4), Thyroid-stimulating hormone (TSH) and thyroid antibodies, thyroid function, duration of orbitopathy, and thyroid disease and therapy of thyroid disease were analysed. The possible existence of risk factors for the development of raised IOP in patients with GO such as family history, smoking, stress, and the presence of systemic diseases was investigated as well.

### Statistical Analyses

The Kolmogorov–Smirnov test assessed continuous numerical value distribution, and non-parametric tests were applied accordingly. Categorical values were presented with frequencies and proportions and analysed using the χ^2^ test or Fisher’s exact test. Median and interquartile ranges depicted continuous values, analysed by the Kruskal–Wallis test for three groups, followed by the post hoc Mann–Whitney U test. Box and Whisker plots illustrated significant differences in s-f changes among GO, GO IOP, and POAG groups. Statistical correlation was interpreted based on the rho coefficient. A multivariate analysis using binary logistic regression was conducted to predict glaucoma occurrence among subjects with orbitopathy. The values obtained from the left eyes were converted and presented as values obtained from the right eyes to ensure mutual comparability across the observed positions. The intensities of functional changes were depicted in their original logarithmic scale of decibels (dB), as non-parametric methods were applied for mutual comparisons and correlations. Significance was set at *p* < 0.05. IBM SPSS Statistics software, version 25.0, was employed for analysis (https://www.ibm.com/analytics/spss-statistics-software (accessed on 23 April 2020)).

## 3. Results

### 3.1. Clinical Features of the Examined Patient Groups

The study included 182 subjects with GO or POAG without signs of other eye diseases divided into three groups. The GO group consisted of 48 subjects (94 eyes) with a clinical presentation of mild to moderate GO and normal values of IOP. The GO IOP group included 50 subjects (97 eyes) with mild to moderate GO and IOP values > 21 mmHg. The POAG group consisted of 84 subjects (153 eyes) with POAG on antiglaucoma therapy. The GO IOP group was, on average, six years older than the subjects in the GO group. The POAG group was, as expected, the oldest, with a ten-year difference compared to the GO group and about five years compared to the GO IOP group (Table 1 and Appendix A). In all three groups, the majority of patients were female. In the GO group 43 (89.6%), in the GO IOP 39 (78%), and in the POAG group 60 (71.4%) were females. The female to male ratio in the GO groups (GO and GO IOP combined) was 5.12:1.

The GO IOP group exhibited a thinner cornea, higher exophthalmometry values, a slightly smaller interquartile range of BCVA and more advanced stages of disease activity and severity (CAS and NOSPECS) compared to the GO group (Table 1 and Table 2). Clinical changes with regard to disease activity parameters in both classifications indicate that the GO IOP group has a significantly higher activity and severity of the disease (Table 2 and Appendix A), although the duration of orbitopathy was on average 2 years in both groups (Appendix A).

### 3.2. Structural–Functional Changes

Figure 2 shows a printout of the Octopus Polar analysis divided into 12 sectors, where the levels of individual functional changes are indicated by the gradation of colours from green to red with regard to their statistical distribution. The damages in dB were summed for each quadrant, and the median was calculated. In the GO group, sectors 11, 12, 1, 6, and 7 had the highest average number of dB, that is, the most pronounced functional changes. In the GO IOP group, a similar distribution prevails, affecting additionally sector 10. The POAG group showed the greatest damage in sectors 6, 7, 11, and 12, and then somewhat less in sector 1 and the entire temporal half (sectors 8–10).

All participants underwent OCT imaging. Measurements included RNFL and other optic nerve head (ONH) parameters, neuroretinal rim thickness (NRR), optic disc surface area, average and vertical cup-to-disc (c/d) ratios, and excavation volume in the three examined groups. The median values of average RNFL thickness are lower in POAG group compared to the two GO groups (*p* < 0.001), with no significant difference observed between them (*p* = 0.907).

Analysis of ONH parameters revealed that the median of NRR was significantly thinner and the average and vertical c/d ratios and cup volume were higher in the GO IOP group compared to the GO group. There was no statistically significant difference in the disc surface area among the three groups. ONH parameters showed the most significant changes in the POAG group, followed by the GO IOP group (Table 3 and Appendix A).

The median values of the RNFL thickness distribution are graphically represented for the optic nerve head, arranged by clock hours from 1 to 12 in the three examined patient groups in Figure 3. The colours indicate the median values of RNFL thickness in microns based on their statistical distribution. The thickness of the sectoral RNFL in the GO and GO IOP groups was within the normal range and there were no significant differences between them. In the POAG group, lower average median values were observed within segments, especially in the superior and inferior quadrants compared to the first two groups.

Structural–functional maps of correlations between s-f changes are shown in Figure 4. Only fields with significant correlations are presented. In the GO group, the strongest correlation was observed in sector 7, in the GO IOP group in sectors 1 and 7, and in the POAG group in sectors 11 to 1 and 6 to 8. While weak negative correlation prevails in the GO and GO IOP groups, in the POAG group several sectors are affected with a medium to strong negative correlation. These maps illustrate the relationship between s-f changes, visually highlighting the optic nerve head and sectors that require attention during clinical examination and analysis of visual field and OCT findings.

### 3.3. Risk Factors for Elevated IOP in Subjects with GO

The thyroid hormone and antibody values, thyroid function, and risk factors were collected at the time of study inclusion and compared between the GO and GO IOP patients. The binary logistic regression model for the prediction of elevated IOP in subjects suffering from GO is shown in Table 4. The regression model was statistically significant (*p* = 0.001), describing 50.3% variance of dependent variable (elevated IOP) and correctly classifying 81.5% of subjects. Of all the risk factors that were part of the regression model, two factors, controlled by the influence of other variables in the regression model, were found as significant predictors of OHT in subjects suffering from orbitopathy: elevated Anti-TG values with a probability ratio (OR) of 21.0 (95% CI 1.59—278.06; *p* = 0.021) and older age with OR = 1.08 (95% CI: 1.01—1.15; *p* = 0.026).

The differences in risk factors between participants in the GO and GO IOP groups showed that smoking was evenly distributed between the two groups (56.3% vs. 54.0%). Stress at the time of thyroid disease diagnosis was significantly more prevalent in the GO group compared to GO IOP (54.2% vs. 30.0%, *p* = 0.015). Most patients in both groups did not have other comorbidities such as diabetes, arterial hypertension, hyperlipidaemia, depression, asthma, or chronic obstructive pulmonary disease (COPD) (Appendix A). Family history was positive for thyroid diseases in about a third of patients in both groups (33.3% in GO and 28% in GO IOP), and in <10% of patients, it was positive for glaucoma or glaucoma and thyroid gland disease (Appendix A).

## 4. Discussion

GO and POAG share some clinical signs such as elevated IOP and visual field damage. Therefore, in patients with GO, the diagnosis of POAG can be challenging. When IOP is identified in these patients, the question arises as to whether it indicates a compressive rise in IOP or if the existing optic nerve changes are attributable to glaucoma or DON [12]. Studies are showing that the prevalence of glaucoma in patients with thyroid pathology is higher compared to the normal population [13,14]. GO is characterized by retraction and swelling of the eyelids, keratitis, proptosis, restrictive myopathy, elevated IOP, decreased visual acuity and, in severe cases, damage to the optic nerve. In the group of patients with POAG, numerous studies have shown characteristic changes in the structure of the optic nerve and macula, followed by the loss of function [7,8,15]. This study aimed to determine the relationship of s-f damage in GO groups with normal and elevated IOP, because in these patients, the data on sectoral optic nerve damage are scarce, and the correlation between sectoral s-f has not yet been investigated [12].

### 4.1. Clinical Features of the Examined Patient Groups

This study investigated the role of IOP in patients with GO, as elevated IOP is a crucial modifiable risk factor influencing the onset and progression of glaucoma. The median IOP values observed in the GO IOP group (24 mmHg) were higher than the POAG group (20 mmHg). Patients with glaucoma were on antiglaucomatous therapy, and hence, the pressure values are somewhat lower than in GO IOP group. Elevated IOP is anticipated in active orbitopathy, potentially contributing to the progression of glaucomatous neuropathy. Known risk factors for glaucoma in the general population can be age, race, elevated IOP, thinner cornea, large c/d ratio, and myopia [10]. This study identified older age, elevated IOP, thinner cornea, higher c/d ratio, increased exophthalmometry values, slightly smaller interquartile range of BCVA and higher disease activity stages (CAS and NOSPECS) as factors associated with elevated IOP in GO patients. Therefore, it can be concluded that most ocular factors for the development of glaucoma were positive in the GO IOP group. Some previous studies investigating IOP values and changes in RNFL thickness and ONH in subjects with GO also found also higher IOP and c/d ratio values and thinner macula and RNFL compared to healthy populations [16,17]. The GO IOP group exhibited a more severe clinical status and suffered from thyroid disease on average one year longer compared to GO group, suggesting a positive association between elevated IOP and the severity of GO. The same factors, high disease activity and duration of GO, for the development of IOP have been recently identified in another study [3,18].

### 4.2. Structural and Functional Changes

The most common perimetric method, the Octopus perimeter, has so far not been sufficiently used in the s-f study in glaucoma or GO. The most applied method in s-f studies, the Humphrey visual field analyser (HFA), is fundamentally different from the Octopus perimeter, and the results obtained cannot be directly applied to the Octopus perimeter [12,19]. Prior studies on Octopus have organised visual field regions based on RNFL pathway anatomy [20,21,22]. The first study compared 16 RNFL sectors and 16 Octopus clusters based on the G2 programme [20]. The other two studies correlated 10 Octopus clusters with peripapillary RNFL thickness in POAG individuals. There were moderate-to-strong relationships for all 10 cluster mean defect and sector peripapillary RNFL thickness pairs, and the strongest relationship was found for the inferotemporal RNFL thickness sector superior and superior paracentral cluster pair [21,22]. Monsalve et al. conducted a similar study in healthy individuals and glaucoma patients, establishing an s-f relationship between visual field areas presented by Octopus and RNFL thickness measured by Spectral Domain OCT in 12 quadrants [23].

Very few studies have explored the s-f relationship in GO patients [17,24]. All of them have examined only the global parameters of the visual field using the HFA [17,24,25].

Forte et al. explored RNFL thickness in four quadrants, optic disc parameters and mean deviation (MD) of HFA visual field in POAG, GO with elevated IOP and healthy subjects, emphasizing higher c/d ratios in POAG versus GO IOP and healthy subjects, along with thicker NRR in the GO IOP group compared to POAG. This study using HFA analyser did not include GO subjects with normal IOP [17]. So far, no study has been published evaluating functional visual field defects in GO, GO IOP, and POAG subjects using the Octopus perimeter and Polar analysis. 

A study in POAG using Polar Trend Analysis (PTA) showed a correlation with structural changes in OCT. The authors demonstrated that in glaucomatous eyes, significant progression in Polar analysis occurred predominantly in the inferior, inferotemporal, superior, and superotemporal peripapillary regions. These locations correspond to the parts of the NRR and peripapillary RNFL known to be most sensitive to glaucoma. The same areas were affected in this study in the POAG group. The study also pointed out that PTA earlier indicated glaucomatous progression compared to linear regression analysis of RNFL and ganglion cell complex (GCC) thickness parameters [26]. The present study found the largest dB losses in the POAG group, followed by the GO IOP group, and then the GO group. This indicates that with the increase in IOP, more damage is present in the visual field, especially in segments corresponding to the upper and lower poles of the optic nerve known to be sensitive in glaucoma [27].

It was recognized before that RNFL thickness is crucial for early diagnosis, as it can be the earliest clinically detectable structural change, occurring up to six years before visual field loss in glaucoma [28]. In GO, analysing RNFL helps identify consequences of orbital congestion, optic neuropathy, and changes related to secondary glaucoma. The overall thickness of the RNFL in patients with GO was lower than normal values in the general population [29,30]. However, the difference in the overall RNFL thickness between the groups with GO and GO IOP was not significant. Other studies also observed RNFL thinning in GO patients without clinical optic nerve dysfunction, particularly in the inferior quadrant [12,16,29]. This suggests the possibility that RNFL damage may occur independently of IOP, and that structural RNFL impairments may exist even in clinically normal findings. Although the sectoral RNFL values were similar in GO and GO IOP groups, the analysis of patients with the highest IOP values (≥28 mmHg) showed that all of them had sectoral RNFL damage. This indicates the positive correlation of the significantly elevated IOP with the structural damage to the optic nerve.

The morphology of the optic disc continues to be a crucial factor in diagnosing various optic neuropathies, such as glaucoma and DON. The average thickness of the NRR was significantly thinner and the average vertical c/d ratios as well as the optic disc volume were significantly larger in the GO IOP group compared to the GO group. These results indicate that these parameters may more effectively detect early changes in the optic nerve head between the GO and GO IOP groups, and better differentiate between them compared to the average RNFL thickness. These results align with a recent study where in 51% of GO patients with raised IOP, the most common optic disc feature was disc cupping [12].

To establish the spatial relationship between RNFL measurements and visual field regions, various s-f maps were created [8,31]. The Octopus software algorithm makes interpretation of the visual field damage possible in such a way that it corresponds to the representation of the optic nerve head on OCT. The analysis of the RNFL thickness in 12 optic nerve sectors and corresponding visual field damage showed that the GO group had a strongest negative correlation in sector 7, while the GO IOP group had correlations in sectors 1 and 7. This confirms the association with the formation or deepening of already present visual field defects, which occur first in zones corresponding to the superior and inferior RNFL fibres. The POAG group exhibited medium–strong negative correlations in sectors 1, 6, 7, 8, 11, and 12. In the earlier stages of glaucoma, localized cell loss occurs specifically in regions such as the superotemporal and inferotemporal peripapillary quadrant. This suggests that they are particularly suitable for glaucoma diagnosis. The same configuration generally corresponds to the pattern obtained on the s-f map in the GO IOP group. The higher IOP values and longer the pressure persists, the existing deficits deepen and spread to additional sectors. It is known that for initial change in visual field, one third of nerve fibres have to be damaged, so we should keep in mind that any sectoral thinning and its extension to the neighbouring sector in two consecutive OCT imaging measurements can be a sign that the damage has progressed [32]. From a clinical point of view, the most important question is when to start treatment of elevated IOP in GO. After numerous studies on glaucoma optic neuropathy [7,8,28,33], it is clear that the functional damage is a sign of already advanced damage at the level of retinal ganglion cells and RNFL. Therefore, the detection of the first changes is a very important step and s-f correlation is a key aspect of glaucoma management in the GO IOP group.

### 4.3. Risk Factors for Elevated IOP in Subjects with GO

The binary logistic regression model was used to obtain a prediction of the development of IOP in subjects suffering from GO. This model included several risk factors and showed that subjects who had elevated Anti-TG values had a 21-times-higher probability of developing IOP elevation. Thyroglobulin (Tg) has been proposed to play a role in GO [34,35,36]. Polymorphisms in Tg, along with a combination of iodine and the enzymatic activity of thyroid peroxidase required for thyroid hormone synthesis, may unintentionally trigger an autoimmune response in the thyroid [36,37,38]. The involvement of Anti-TG in GO pathogenesis and their role as risk factor for IOP remains to be elucidated. If Anti-TG contributes to GO etiopathogenesis, it is unlikely to be the sole factor, and its presence in orbital tissues may exacerbate clinical symptoms initiated by other mechanisms, potentially directly affecting IOP [39,40].

Another significant variable is age—for each year of age, the probability of developing elevated IOP rises 1.08 times, or 8%, confirming age as a risk factor for elevated IOP. In a study conducted on 1280 participants with GO, OHT was statistically significantly associated with older age (≥45 years) [3]. In older age, thyroid disease tends to have a longer duration, and thyroid hormones and antibodies can impact the eyes over an extended period [39,40]. 

Both GO (56.3%) and GO IOP (54%) groups had a higher percentage of smokers than non-smokers, exceeding the smoking rate in Croatia [41]. However, as the difference between the two groups was not significant, it cannot be claimed that smoking is associated with elevated IOP values and optic neuropathy. Stress at the time of thyroid disease diagnosis was more prevalent in the GO group compared to GO IOP, suggesting a potential role in triggering GO, although it may not influence the development of elevated IOP.

## 5. Conclusions

This study is the first to utilize Polar analysis, revealing s-f changes in patients with GO and POAG. Through this examination, even subtle changes in the visual field that may not appear significant in conventional visual field imaging can be identified and further analysed. We presented s-f maps in glaucoma and GO, as well as risk factors influencing the development of IOP in GO. 

Our study had some limitations due to its cross-sectional design, and therefore, it is restricted in its ability to demonstrate long-term progression and the incidence of glaucoma in patients with GO. Another limitation of this study is the inability to distinguish between POAG and a secondary increase in IOP resulting from other forms of compressive optic neuropathy in GO patients. However, through detailed patient medical history and radiological evaluations, a highly effective patient selection process was achieved. It would be more accurate to measure the GCC and more precisely categorize patients based on age. 

The study suggests that monitoring IOP and performing annual visual field and OCT tests is crucial in GO patients, with antiglaucoma therapy initiation upon elevated IOP detection before further structural changes occur. Treatment should be maintained during active disease stages (CAS above 2–3), with periodic reassessment after disease activity subsides. Finally, further longitudinal studies with larger samples should be performed in future.

## Figures and Tables

**Figure 1 diagnostics-14-00649-f001:**
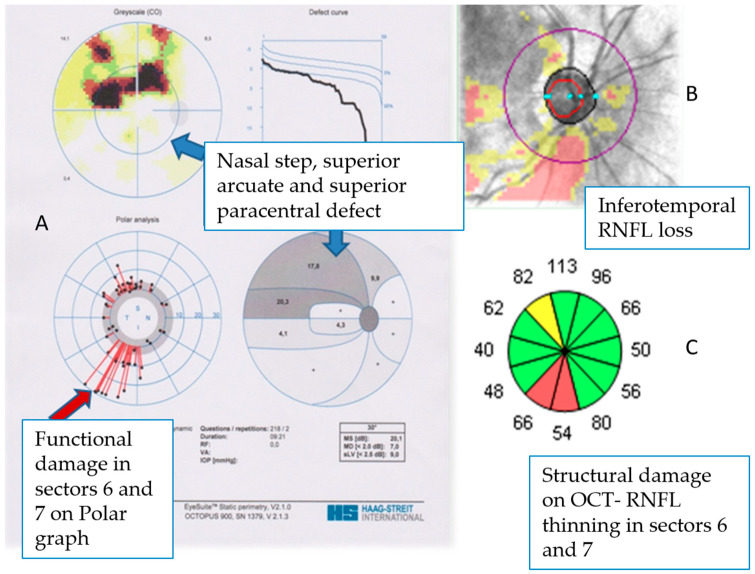
Representation of the relationship between structural and functional changes. (**A**) The Octopus EyeSuite software 4.4.1 (Haag−Streit International, Köniz, Switzerland) provided Polar analysis. Localization of the red lines indicates the location of the damage, and the length of the line indicates the extent of the loss in dB; the longer the line, the greater the damage. The orientation of the graph is mirror − reversed to correspond to the anatomy of the optic nerve head. (**B**) The OCT RNFL deviation map with visible inferotemporal RNFL loss. (**C**) On the RNFL clock hours map RNFL thinning is visible in the same quadrants as on the Polar graph.

**Figure 2 diagnostics-14-00649-f002:**
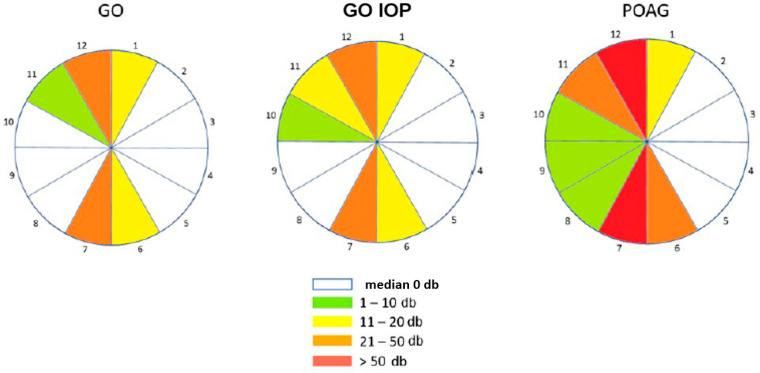
The median of functional changes in dB in 12 quadrants in 3 groups are visually presented: GO—Graves’ orbitopathy; GO IOP—Graves’ orbitopathy with elevated intraocular pressure; POAG—primary open-angle glaucoma.

**Figure 3 diagnostics-14-00649-f003:**
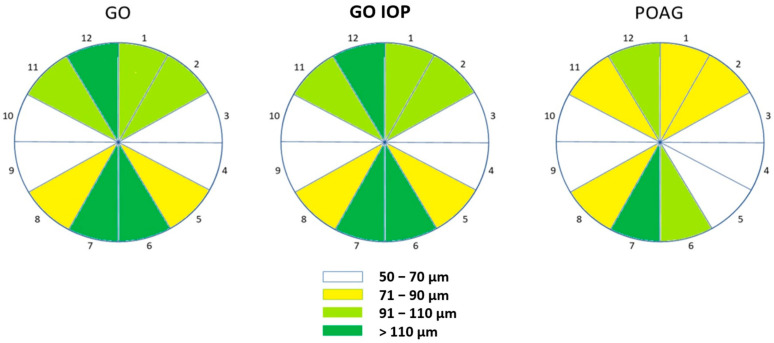
The graphic representation illustrates the median values of RNFL thickness distribution in microns (μm) for the optic nerve head. The data are organized by clock hours, ranging from 1 to 12, across the three examined patient groups. There is no significant difference between GO and GO IOP groups that have RNFL thickness within normal range. In POAG group, lower average median RNFL thickness values were observed across segments. The legend shows the median RNFL thickness compared to the normal range for each individual sector of the optic nerve head. GO—Graves’ orbitopathy; GO IOP—Graves’ orbitopathy with elevated intraocular pressure; POAG—primary open-angle glaucoma.

**Figure 4 diagnostics-14-00649-f004:**
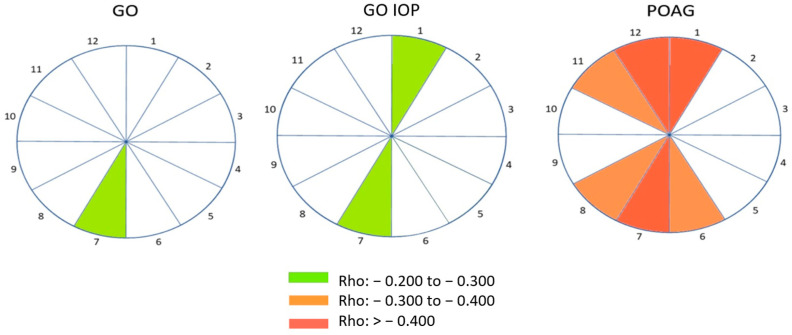
Structural–functional maps of significant correlations between s-f changes in 12 optic nerve head sectors in GO, GO IOP, and POAG patients’ group. The Spearman correlation coefficients (rho) were used in the analysis of the correlation between individual s-f changes for each group. A statistically significant correlation (*p* < 0.05) was interpreted as follows: for the absolute value of the rho coefficient from 0 to 0.3, weak correlation; for rho from 0.3 to 0.4, moderately strong correlation; for rho greater than 0.4, very good to excellent correlation. GO—Graves’ orbitopathy; GO IOP—Graves’ orbitopathy with elevated intraocular pressure; POAG—primary open-angle glaucoma; Rho: the Spearman correlation coefficient.

**Table 1 diagnostics-14-00649-t001:** Descriptive statistics of examined clinical parameters and comparison between GO, GO IOP, and POAG groups: Kruskal–Wallis test. GO—Graves’ orbitopathy; GO IOP—Graves’ orbitopathy with elevated intraocular pressure; POAG—primary open-angle glaucoma BCVA—best corrected visual acuity; IOP—intraocular pressure; CCT—central corneal thickness.

Group		N	Min	Max	Centile			*p*
					25.	Median	75.	
Age (yrs)	GO	48	22.00	81.00	45.50	55.50	63.50	<0.001
	GO IOP	50	28.00	77.00	52.50	61.50	68.25	
	POAG	84	18.00	90.00	57.25	65.00	73.00	
BCVA	GO	94	0.80	1.00	1.00	1.00	1.00	<0.001
	GO IOP	97	0.80	1.00	0.90	1.00	1.00	
	POAG	153	0.80	1.00	0.80	1.00	1.00	
IOP (mmHg)	GO	94	10.00	20.00	14.00	16.00	17.00	<0.001
	GO IOP	97	21.00	28.00	22.00	24.00	26.00	
	POAG	153	10.00	26.00	18.00	20.00	22.50	
CCT (μm)	GO	94	465.00	617.00	531.75	555.50	575.25	<0.001
	GO IOP	97	433.00	658.00	521.00	541.00	563.50	
	POAG	153	450.00	669.00	506.00	535.00	558.50	
Exophthalmometry (mm)	GO	94	10.00	27.00	16.00	18.00	21.00	<0.001
	GO IOP	97	11.00	29.00	17.00	20.00	23.00	
	POAG	153	13.00	19.00	15.00	16.00	17.00	

**Table 2 diagnostics-14-00649-t002:** Differences in clinical activity and extension of changes with respect to disease activity parameters evaluated by CAS and NOSPECS classification: Kruskal–Wallis test. GO—Graves’ orbitopathy; GO IOP—Graves’ orbitopathy with elevated intraocular pressure; CAS—Clinical Activity Score; NOSPECS criteria used to evaluate the severity of GO.

Group		N	Min	Max	Centile			*p*
					25.	Median	75.	
CAS	GO	94	0.00	3.00	0.00	0.00	1.00	<0.001
	GO IOP	97	0.00	5.00	1.00	1.00	3.00	
NOSPECS classification	GO	94	0.00	3.00	0.00	1.00	2.00	<0.001
	GO IOP	97	0.00	3.00	1.00	2.00	3.00	

**Table 3 diagnostics-14-00649-t003:** RNFL thickness and ONH parameters measured using OCT in the three examined groups. GO—Graves’ orbitopathy; GO IOP—Graves’ orbitopathy with elevated intraocular pressure; POAG—primary open-angle; RNFL—retinal nerve fibre layer.

Group		N	Min	Max	Centile			*p*
					25.	Median	75.	
RNFL (μm)	GO	94	61.00	113.00	83.00	89.50	96.25	<0.001
	GO IOP	97	66.00	110.00	85.00	89.00	94.00	
	POAG	153	49.00	111.00	63.00	80.00	89.00	
Rim area (mm^2^)	GO	94	1.00	2.13	1.30	1.51	1.66	<0.001
	GO IOP	97	0.87	2.09	1.20	1.35	1.53	
	POAG	153	0.21	2.80	0.87	1.09	1.32	
Disc area (mm^2^)	GO	94	1.12	2.85	1.66	1.85	2.08	0.327
	GO IOP	97	1.27	2.83	1.68	1.96	2.17	
	POAG	153	0.78	3.25	1.59	1.90	2.24	
Average c/d ratio	GO	94	0.07	0.70	0.30	0.40	0.55	<0.001
	GO IOP	97	0.12	0.71	0.40	0.50	0.62	
	POAG	153	0.05	0.93	0.56	0.65	0.74	
Vertical c/d ratio	GO	94	0.06	0.66	0.30	0.40	0.51	<0.001
	GO IOP	97	0.09	0.74	0.41	0.50	0.60	
	POAG	153	0.05	0.91	0.55	0.62	0.75	
Cup volume (mm^3^)	GO	94	0.00	0.66	0.02	0.06	0.15	<0.001
	GO IOP	97	0.00	0.51	0.03	0.09	0.23	
	POAG	153	0.00	1.43	0.14	0.25	0.40	

**Table 4 diagnostics-14-00649-t004:** Binary logistic regression model of prediction of elevated IOP in subjects suffering from orbitopathy; * Significant risk factors; TSH—thyroid-stimulating hormone, thyrotropin. fT3—free fraction of triiodothyronine; fT4—free fraction of thyroxine. Anti-TSHR—antibodies against TSH receptors; Anti-TPO—microsomal antibodies against thyroid peroxidase; Anti-TG—antibodies against thyroglobulin.

r^2^ = 50.3%; *p* = 0.001	OR	95% CI		*p*
		Low	High	
Age (yrs)	1.08	1.01	1.15	0.026 *
Female sex	2.96	0.51	17.19	0.226
Smoking	2.40	0.47	12.30	0.294
Arterial Hypertension	3.51	0.55	22.29	0.183
DM	1.36	0.11	16.57	0.812
Stress at the time of diagnosis of thyroid disease	0.90	0.18	4.37	0.892
Thyroid function on diagnosis of GO: hipo				0.305
Thyroid function on diagnosis of GO: eu	6.96	0.57	84.56	0.128
Thyroid function on diagnosis of GO: hiper	2.97	0.36	24.14	0.309
TSH < 0.35 mIU/L				0.498
TSH 0.35–4.94 mIU/L	2.62	0.26	25.96	0.411
TSH > 4.94 mIU/L	0.77	0.04	14.84	0.860
fT3 < 3.8 pmol/L				0.141
FT3 3.8–6.0 pmol/L	0.57	0.12	2.69	0.481
FT3 > 6 pmol/L	22.95	0.89	591.99	0.059
fT4 < 9.0 pmol/L				0.114
FT4 9.0–19.0 pmol/L	0.00	0.00		0.999
FT4 > 19.0 pmol/L	0.00	0.00		0.999
Anti-TSHR elevated values	0.41	0.11	1.52	0.184
Anti-TPO elevated values	1.46	0.26	8.13	0.663
Anti-TG elevated values	21.00	1.59	278.06	0.021 *

## Data Availability

The raw data supporting the conclusions of this article will be made available by the authors on request.

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
