# Peer review of "Correlation between Structural and Functional Changes in Patients with Raised Intraocular Pressure Due to Graves’ Orbitopathy"

_diagnostics, 2024, doi:10.3390/diagnostics14060649_

Round 1

Reviewer 1 Report

Comments and Suggestions for Authors

Authors utilized Octopus 900 visual field testing and OCT to establish correlations between functional and structural changes in optic nerve regions in patients with GO and patients with GO with elevated IOP (GO IOP) groups. The idea is novel and important; however, there are some issues to be addressed.

1.       According to the data, there are differences in the onset age of thyroid disease and GO between the GO and GO IOP groups. Is there a correlation between disease severity and age?

2.       The authors identified age as a risk factor for IOP elevation. Could it actually be related to the severity of the disease?

3.       In Tables 1 and 5, could the authors provide the P value between GO and GO IOP as well? This would be more straightforward.

4.       Are there any reasons for the absence of Tables 3 and 4?

5.       In Figure 4, structural-functional maps of significant correlations between s-f changes in 12 optic nerve head sectors in GO and GO IOP groups showed weaker correlation compared with the POAG patients' group. What could explain this difference?

Author Response

Dear sir,

Article: Correlation between structural and functional changes in patients with raised intraocular pressure due to Graves' orbitopathy

Response to Reviewer 1 Comments

Dear reviewer 1,

Thank you very much for taking the necessary time and effort to review this manuscript. I sincerely appreciate all your valuable comments and suggestions, which helped us in improving the quality of the manuscript. Please find the detailed responses to your comments and suggestions below. The corresponding corrections are highlighted in red.

Comment 1: According to the data, there are differences in the onset age of thyroid disease and GO between the GO and GO IOP groups. Is there a correlation between disease severity and age?

Response 1: The GO IOP group exhibited a more severe clinical status according to CAS and NOSPECS classifications (Table 2 and Supplementary Table S2) and were on average 5 years older compared to the GO group (Table 1). Additionally, they suffered from thyroid disease on average one year longer compared to GO group (Supplementary Table S3). All this indicates that there might be a correlation between disease severity and age.

Comment 2: The authors identified age as a risk factor for IOP elevation. Could it actually be related to the severity of the disease?

Response 2: In the GO group with elevated IOP, higher stages of disease activity and severity were recorded and those patients were approximately 5 yrs older than patients in GO group with normal IOP. With the progressing age, the severity of the disease is progressing. So both, age and severity of the disease are positively correlated to the raised IOP.

Comment 3: In Tables 1 and 5, could the authors provide the P value between GO and GO IOP as well? This would be more straightforward.

Response 3: Please find below the requested addition to the tables. We have added this in Suppl. Table 1 and 4.

Table 1 Post-hoc analysis of differences in the examined clinical parameters: Mann-Whitney U test.

Post-hoc comparisons

P value

GO vs GO IOP

GO vs POAG

GO IOP vs POAG

IOP (mmHg)

<0,001

<0,001

0,001

CCT (μm)

0,018

<0,001

0,086

Exophthalmometry (mm)

0,002

<0,001

<0,001

BCVA

0,007

<0,001

0,005

GO - Graves' orbitopathy, GO IOP - Graves' orbitopathy with elevated intraocular pressure, POAG- primary open-angle glaucoma; IOP- intraocular pressure; CCT- central corneal thickness; BCVA - best corrected visual acuity

Table 2. Post-hoc testing for differences in clinical changes with respect to optic nerve parameters: Mann-Whitney U test.

Post-hoc comparisons

P value

GO vs GO IOP

GO vs POAG

GO IOP vs POAG

RNFL (μm)

0,907

<0,001

<0,001

Rim area (mm2)

0,001

<0,001

<0,001

Disc area (mm2)

0,145

0,218

0,989

Average c/d ratio

<0,001

<0,001

<0,001

Vertical c/d ratio

<0,001

<0,001

<0,001

Cup  volume (mm3)

0,007

<0,001

<0,001

GO - Graves' orbitopathy, GO IOP - Graves' orbitopathy with elevated intraocular pressure, POAG- primary open-angle, RNFL- retinal nerve fibre layer, c/d - cup/disc

Comment 4: Are there any reasons for the absence of Tables 3 and 4?

Response 4: When we moved the tables 2 and 3 into the supplement, we overlooked changing the table 5 number. It is now corrected.

Comment 5: In Figure 4, structural-functional maps of significant correlations between s-f changes in 12 optic nerve head sectors in GO and GO IOP groups showed weaker correlation compared with the POAG patients' group. What could explain this difference?

Response 4: Several factors could explain the weaker correlation observed in the structural-functional maps of significant correlations between structural and functional changes in the 12 optic nerve head sectors in the GO and GO IOP groups compared to the POAG patients' group.

We included glaucoma patients in the study because glaucoma provides the best model for researching structural and functional changes and their interconnection. Glaucoma is a disease where to establish a diagnosis, there must be both structural and functional damage. In the glaucoma group, there were more pronounced changes in functional and structural parameters, hence the correlation was stronger.

In the groups with orbitopathy, we included patients with mild and moderate stages and divided them into two groups based on elevated intraocular pressure. Since we included patients as soon as their elevated intraocular pressure was measured, there were no significant structural damages because the pressure elevation did not last long. However, by analyzing individual cases of patients who had higher IOP values (≥28 mmHg), it was found that all of them had sectoral RNFL damage, which clearly indicates that the more pronounced elevated IOP values are associated with structural damage to the optic nerve.

Some other potential explanations could include differences in the pathophysiology of POAG and secondary IOP raise in GO, variations in disease progression rates, suggestion that SAP is relatively insensitive to early changes in the visual field, disparities in treatment responses, or distinct mechanisms underlying optic nerve damage and dysfunction of retinal ganglion cells. Additionally, differences in patient characteristics such as age, comorbidities, or disease duration could also contribute to this discrepancy.

The concept of 'dysfunction of ganglion cells' rather than their death can explain why functional deficits precede structural changes in some patients. In the early stages of damage, ganglion cells may become dysfunctional, leading to reduced sensitivity in the visual field, even though the thickness of RGCs and their axons is preserved and that could be the case in GO groups.

Kind regards,

Dr. Freja Bagatin

Reviewer 2 Report

Comments and Suggestions for Authors
  1. In lines 8 and 32, I don't think that elevation of IOP in thyroid eye disease (TED) is rare.
  2. In line 26, kindly delete the comma prior to "is".
  3. Did the authors measure IOP in the upward gaze? What was the gaze during IOP measurement in patients with TED-related strabismus?
  4. In line 32, kindly remove “a” prior to “rare”. 
  5. Was MRI requested to check TED activity? MRI is more reliable to determine TED activity than CAS.
  6. Why only include mild to moderate POAG? 
  7. How was “stress” measured?
  8. In line 70, kindy remove “)” after gonioscopy.
  9. What determined the 1 month interval between visual field testing?
  10. In line 88, please add a comma after EUGOGO classification.
  11. I recommend the authors to move the subsection of statistical analyses (3.1) into the Methods section.
  12. Please fix the first sentence in line 120.  Perhaps a conjunction before “divided” will provide better readability. The manuscript has several run-on sentences like that in lines 259-262, 339-340, etc. Please check and edit these. 
  13. Authors can abbreviate “primary open angle glaucoma” in line 124.
  14. In line 128, the authors mentioned that age is a risk factor for elevated IOP, but they only presented the difference in age among the groups here. I think that this presentation is not enough to conclude that age is a risk factor for elevated IOP in TED.
  15. Please clarify the sentence in line 129. Are the numbers referring to the count of female subjects?
  16. In line 131, change “IOT” to “IOP”.
  17. In lines 139-140, NOSPECS is a scale for severity of TED, not activity.
  18. Please fix table 5, “110,00” to “110.00”.
  19. Kindly use superscript numbers of "2" and "3" after "mm" in Table 5.
  20. Lines 191-192 in the results are better placed in the discussion.
  21. In line 208, add a comma after POAG group.
  22. In line 143, average duration of orbitopathy was 2 years for both GO and GO IOP groups.  In line 279, GO IOP group suffered from thyroid disease one year longer compared to GO group. Please clarify this.
  23. No need to abbreviate “retinal ganglion cells” in line 360, “thyroid peroxidase” in line 369 since these abbreviations are not used after the previously mentioned lines.
  24. In line 380, please edit “GO GO IOP”.
  25. In Table 6, please clarify the legends in the caption (eg. caption says Anti-TPO, but the table shows TPO).
Comments on the Quality of English Language

I give my comment above.

Author Response

Dear sir,

Article: Correlation between structural and functional changes in patients with raised intraocular pressure due to Graves' orbitopathy

Response to Reviewer 2 Comments

Dear reviewer 2,

Thank you very much for taking the necessary time and effort to review this manuscript. I sincerely appreciate all your valuable comments and suggestions, which helped us in improving the quality of the manuscript. Please find the detailed responses to your comments and suggestions below. The corresponding corrections are highlighted in red.

Comment 1: In lines 8 and 32, I don't think that elevation of IOP in thyroid eye disease (TED) is rare.

Response 1: Studies report a prevalence of elevated IOP in GO ranging from 3.1% to 24%. The prevalence of ocular hypertension (OH) in GO patients is reported to be 3.1%, and the prevalence of glaucoma is 2.8%. We considered that those reports indicate that the prevalence is rare (PMID: 33564472; PMID: 35615845; PMID: 31440023; PMID: 9924366. PMID: 18535600; PMID: 29557835). Now we have corrected this in the above-mentioned sentences deleting the term „rare“.

Comment 2: In line 26, kindly delete the comma prior to "is".

Response 2: Done

Comment 3: Did the authors measure IOP in the upward gaze? What was the gaze during IOP measurement in patients with TED-related strabismus?

Response 3: IOP was measured in the primary position, with the subject not moving the bulbi upward, as this could potentially elevate IOP.

Comment 4: In line 32, kindly remove “a” prior to “rare”. 

Response 4: Done

Comment 5: Was MRI requested to check TED activity? MRI is more reliabl to determine TED activity than CAS.

Response 5: The diagnosis of GO was made based on clinical presentation, characteristic findings on computed tomography, laboratory, and endocrinological findings.

During the study, we used CT in the diagnostic algorithm because MR is a lengthy, difficult to access, and expensive method. Only some patients were considered for MRI. We concluded that the CT with the assessment of the activity and severity through standardized classifications are adequate approach for this research.

Comment 6: Why only include mild to moderate POAG? 

Response 6: We aimed to include patients with better central visual acuity and mild to moderate stages to better describe the relationship between structure and function. In advanced glaucoma, due to the floor effect, this correlation could not be accurately determined. Additionally, since we compared them with patients with orbitopathy, we wanted comparable visual field stages.

Comment 7: How was “stress” measured?

Response 7: The stress was recorded based on the self-reported interview.

Comment 8: In line 70, kindy remove “)” after gonioscopy.

Response 8: Done

Comment 9: What determined the 1 month interval between visual field testing?

Response 9: We opted for a one-month interval as it was close enough not to affect the test results and it was convenient for the patients to come for visual field testing within that timeframe.

Comment 10: In line 88, please add a comma after EUGOGO classification.

Response 10: Done

Comment 11: I recommend the authors to move the subsection of statistical analyses (3.1) into the Methods section.

Response 11: I agree. It is done.

Comment 12: Please fix the first sentence in line 120.  Perhaps a conjunction before “divided” will provide better readability. The manuscript has several run-on sentences like that in lines 259-262, 339-340, etc. Please check and edit these. 

Response 12: The sentence in mine line 120 is this one: „Significance was set at P < 0.05. IBM SPSS Statistics software, version 25.0, was employed for analysis (https://www.ibm.com/analytics/spss-statistics-software)“ . So, there must be some mistake in line number. If you would be so kind to quote the entire sentence. I will be happy to correct.

Lines 259-262: The sentence is rewritten.

Line 339-340: The sentence is corrected if this is the right one.

Comment 13: Authors can abbreviate “primary open angle glaucoma” in line 124.

Response 13: Done

Comment 14: In line 128, the authors mentioned that age is a risk factor for elevated IOP, but they only presented the difference in age among the groups here. I think that this presentation is not enough to conclude that age is a risk factor for elevated IOP in TED.

Response 14: Patients who had elevated intraocular pressure in GO were statistically significantly older compared to those who had normal intraocular pressure values. Therefore we concluded that age is a relative risk factor. Same was confirmed by binary logistic regression model.

Comment 15: Please clarify the sentence in line 129. Are the numbers referring to the count of female subjects?

Response 15: In the GO group 43 (89.6%), in the GO IOP 39 (78%) and in the POAG group 60 (71.4%) were females. This is now clarified in text.

Comment 16: In line 131, change “IOT” to “IOP”.

Response 16: Done

Comment 17: In lines 139-140, NOSPECS is a scale for severity of TED, not activity.

Response 17: Added and corrected.

Comment 18: Please fix table 5, “110,00” to “110.00”.

Response 18: Corrected

Comment 19: Kindly use superscript numbers of "2" and "3" after "mm" in Table 5.

Response 19: Corrected

Comment 20: Lines 191-192 in the results are better placed in the discussion.

Response 20: Here again I am not sure that the lines you mentioned correspond to the my version of the manuscript. Could you please quote the sentences that are better suited in the discussion and I will make changes accordingly.

Comment 21: In line 208, add a comma after POAG group.

Response 21: Done

Comment 22: In line 143, average duration of orbitopathy was 2 years for both GO and GO IOP groups.  In line 279, GO IOP group suffered from thyroid disease one year longer compared to GO group. Please clarify this.

Response 22: In two groups of subjects with GO, the duration of thyroid disease was examined, which was 4 years in the GO group and 5 years in the GO IOP group. Although not statistically significant (P=0.431), participants with elevated IOP had, on average, one more year of thyroid disease. The duration of orbitopathy was approximately the same in both groups, around 2 years (P=0.435) (Table S3).

Group

N

Min

Max

Centile

P

25.

Median

75.

Duration of thyroid disease (yrs)

GO

48

0.50

38,00

2.00

4.00

10.00

0.431

GO IOP

50

0.50

30.00

2.88

5.00

13.00

Duration of orbitopathy (yrs)

GO

48

0.50

13.00

1.00

2.00

3.00

0.435

GO IOP

50

0.00

19.00

1.00

2.00

4.00

Comment 23: No need to abbreviate “retinal ganglion cells” in line 360, “thyroid peroxidase” in line 369 since these abbreviations are not used after the previously mentioned lines.

Response 23: Agreed and corrected.

Comment 24: In line 380, please edit “GO GO IOP”.

Response 24: Done

Comment 25: In Table 6, please clarify the legends in the caption (eg. caption says Anti-TPO, but the table shows TPO).

Response 25: Thank you for pointing this out. Table is now corrected.

Kind regards,

Dr. Freja Bagatin

Round 2

Reviewer 2 Report

Comments and Suggestions for Authors

1. In line 35, "is" should be "are". To add, our comment was that these complications are not rare. They just changed "rare" to "not common", which are basically the same. 

2. I know that the authors confirmed age as a risk factor using logistic regression model. But in line 133, they only presented the difference in age among the groups here. Therefore, this description is not adequate and should be moved to the discussion section.

3. The sentence "this suggests that superior and inferior quadrants are particularly suitable for glaucoma diagnosis" in lines 199-200 should be moved to the discussion section because suggestions are not presentation of study results.

4. The format of Table 1 should be fixed so that the word "POAG" is in one line.

5. Table 5 has been omitted. But some tables used 'decimal comma' and some used 'decimal point'. I think it would be better if they are consistent with this throughout the manuscript. 6. Sentence in line 353 to 355 is still a run-on sentence. Comma should be placed between "IOP" and "the".

Comments on the Quality of English Language

I mentioned above.
